

# Integrated evaluation of water-related disasters using the analytical hierarchy process under land use change and climate change issues in Laos

Phrakonkham Sengphrachanh[1], Kazama So[1], Komori Daisuke[1]

[1]Department of Civil Engineering, Tohoku University, Sendai, 980-8579, Japan

*Correspondence to*: Sengphrachanh Phrakonkham (phrakonkham.sengphrachanh.t7@dc.tohoku.ac.jp)

**Abstract.** In the past few decades, various natural hazards have occurred in Laos. To lower the consequences and losses caused by hazardous events, it is important to understand the magnitude of each hazard and the potential impact area. The main objective of this study was to propose a new approach to integrating hazard maps to detect hazardous areas on a national scale, for which area-limited data are available. The integrated hazard maps were based on a merging of five hazard maps: floods, land use changes, landslides, climate change impacts on floods and climate change impacts on landslides. The integrated hazard map consists of 6 maps under 3 representative concentration pathway (RCP) scenarios and 2 time periods (near future and far future). The analytical hierarchy process (AHP) was used as a tool to combine the different hazard maps into an integrated hazard map. From the results, comparing the increase in the very high-hazard area between the integrated hazard maps of the far future under the RCP2.6 and RCP4.5 scenarios, Khammouan Province has the highest increase (16.45%). Additionally, the very high-hazard area in Khammouan Province increased by approximately 12.47% between the integrated hazard maps under the RCP4.5 and 8.5 scenarios of the far future.

## 1 INTRODUCTION

Lao People's Democratic Republic or Lao PDR is a developing country located in Southeast Asia. The citizens depend heavily on agriculture and natural resources for their livelihoods. Currently, the water supply system in the country is not well distributed, particularly in rural areas. Therefore, most people living in rural areas are resettled downstream of dams and irrigation areas (Baird and Shoemaker, 2007). Changes in land use, such as decreases in forest density, can lead to increases in flood magnitude (Jongman et al., 2012; Winsemius et al., 2016). In recent years, many researchers have conducted global studies on the impact of climate change on the water cycle and its effect on people's livelihoods (Adeloye et al., 2013; Parmesan and Yohe, 2003; Westra et al., 2014). However, there have been only a few assessments and analyses for predictions on the environmental impacts on the country when considering possible climate changes. According to the Intergovernmental Panel on Climate Change (IPCC) report, Southeast Asia will suffer from increasing flood frequency in the future (IPCC, 2007) general circulation models (GCMs) have been developed to study future climate scenarios and the associated impacts, and they help support strategies and mitigation plans to address the effect of climate change.



The effects of hazards on an area could be in either a single or multiple form. In the last decade, the uses of multihazard assessment focusing on all scales have been considered in several studies (Cutter et al., 2000; Marzocchi et al., 2012; Sendai Framework, 2015; Sullivan-Wiley and Short Gianotti, 2017). However, exhaustive data are required in most assessments. Recently, geographic information systems (GIS) have been used as a tool for such assessment (Fernández and Lutz, 2010; Kazakis et al., 2015). In contrast, the tool is ineffective in performing multicriteria analyses, and hence, it is not appropriate

for executive or managerial purposes. Previous studies have presented many methodologies to integrate multiple hazards, such as using classification schemes or providing weighting for each hazard. There are several multicriteria decision-making methods to solve multiple conflicts among independent criteria when evaluating multihazard maps. For instance, multiattribute utility theory (MAUT) (Keeney and Raiffa, 1993) can decide the best course of action in a given problem by assigning a utility to every possible consequence and calculating the best possible utility. The drawback of this method is the

requirement of a large amount of input in every step of the procedure (Konidari and Mavrakis, 2007). Simple additive weight (SAW) (Fishburn, 1967) was established based on a simple addition of scores that represent the goal achievement under each criterion, multiplied by the particular weight. The disadvantage of SAW is that the estimated weight does not always reflect the real situation (Qin et al., 2008). The technique for order preference by similarity to ideal solutions (TOSIS) (Hwang and Yoon, 1981) is an approach to identify an alternative that is close to an ideal solution and farthest from a nonideal solution in

a multidimensional space. The drawback of this method is the difficulty of weighting criteria and maintaining consistent judgment, especially with additional criteria (Behzadian et al., 2012).

However, none of the studies have taken into consideration the natural abilities of humans to sense, adapt, or modify their environment to avoid danger, which is the human perception of risk as individuals and the public perception of risk as communities or groups. Stakeholder involvement in the study will provide advantages to both researchers and stakeholders.

The stakeholders will have opportunities to share their visions, needs and knowledge on the hazards. They could also assist in reducing conflicts and increasing cooperation in the future. One of the most common Multi Criterial Decision Analysis (MCDA) methods is the analytic hierarchy process. The analytical hierarchy process (AHP) (Saaty, 1994) uses a pairwise comparison to compare the relative significance among criteria designed from the stakeholder's judgment. Although AHP requires data to properly perform pairwise comparisons, it is not nearly as data intensive as MAUT (Kazakis et al., 2015;

Stefanidis and Stathis, 2013). Among various multicriteria decision-making methods, the property of the AHP is in line with our study objective. Furthermore, AHP is recognized as a multicriteria method that is incorporated into GIS-based procedures for determining suitability (Parry et al., 2018; Prakash, 2003). Pourkhabbaz et al. (2014) used AHP in a GIS environment with the aim of choosing a suitable location for agricultural land use. Gigović et al. (2017) presented a reliable GIS-AHP methodology for hazard zone mapping of flood-prone areas in urban areas. From the results, the GIS-AHP hazard

map provides good correlation between the high hazard area of the map and historical flood events. Ramya et al. (2019) analyzed suitable locations for industrial development by using GIS, AHP and the Technique for Order Preference by Similarity to Ideal Solution (TOSIP). As a result, the most suitable industrial locations can be highlighted. Based on the research studies mentioned above, it could be concluded that the AHP is an effective and powerful tool to analyze, structure





and prioritize complex problems considering expert judgment on various aspects. Therefore, the AHP is chosen for the studies of integrated multihazard risk mapping.

The main objective of this study is to propose a reliable hazard map that can identify sensitive areas over the national region, for which limited data are available. This method of modeling combines different hazard maps, including flood, land use change and climate change maps. The proposed methodology provides an integrated hazard map that can be used as a guide map that provides all of the important information that can be used to develop countermeasures not only for floods but also for other natural hazards. This study is also the first to develop a hazard map for the entire country of Laos. Another advantage of this proposed method is that the AHP weights that are used to develop the unified hazard maps are based on the design criteria and priorities of the decision makers. It is helpful for identifying hazard areas and focusing on potential areas of impact.

## 2 STUDY AREA AND DATA

The Lao PDR, or Laos, is situated in the middle of Southeast Asia. The country is landlocked, so it has no direct access to the sea and has common borders with China, Vietnam, Cambodia, Thailand and Myanmar. The country is located in the center of the Indochinese peninsula, located between longitude 100 to 108 degrees east and latitude 14 to 23 degrees north, with a total area of 236,800 km$^2$. The Mekong River flows through almost 1,900 km of Lao territory from the north to the south, and it forms a natural border with Thailand over 800 km. In addition, Lao PDR can be divided into 3 regions. These regions are determined by the Lao government, namely, the southern, central and northern regions (Figure s1 from the supplemental material). Furthermore, Lao PDR is divided into 16 provinces and one capital, Vientiane Capital (Figure s2).

For this study, we used hydrological and meteorological datasets from (Phrakonkham et al., 2019). The rainfall data were interpolated to a 1 km × 1 km resolution using inverse distance weight (IDW). After that, the log-Pearson type III distribution was used to estimate the 100-year return period of extreme rainfall in Laos by using the annual maximum daily rainfall for each grid area. The hydrological data were used as input data for the rainfall-runoff model and probability of landslide model, and to calibrate the rainfall-runoff model. The land use of Laos is classified into forest, paddy field, agricultural area, water body and urban.

In this study, a 100-year return period is used because most of the hazardous events have occurred due to the 100-year return period of extreme rainfall. In addition to the rainfall data, daily maximum data are selected to analyze the rainfall intensity return period. The data were also used for bias correction between GCMs and observation data. In this study, Representative Concentration Pathway (RCP) scenarios were used for future climate change projections because RCP scenario areas based on radiative forcing projections are allowed for policy change to be implemented. Seven GCMs, namely, CanESM2, CNRM-CM5, GFDL-ESM2 M, MPI_ESM_LR, MRI-CGCM3, Miroc-ESM and Miroc-ESM-CHEM (details about each GCM are shown in Table s1), were selected to create future scenarios of spatially distributed heavy rainfall. Rainfall data from the GCMs have different time resolutions; therefore, we converted all 3 h rainfall data to daily





data by summing rainfall data from the same day. The rainfall data period was from 2006 to 2100, and three RCPs were used, including RCP2.6, RCP4.5 and RCP8.5..

## 3 Methodology

### 3.1 Outline of method

In this study, the integrated hazard maps consist of five hazard maps: floods, land use changes, landslides, climate change impacts on floods and climate change impacts on landslides.

### 3.2 Flood hazards

In this study, the model considers the meteorological dataset as input into an output hydrological dataset such as streamflow over a time period. A hydrological model is made of mathematical representations of key processes, such as precipitation,
infiltration and transfer into streams; the hydrological processes considered in this model are precipitation, infiltration, surface runoff, base water flow and water balance in each layer. The model technically consists of a set of hydrological parameters describing the catchment properties and algorithms describing the physical processes. In this model, the catchment is divided into overland flow planes and channel segments. On the land, for each grid cell, two layers are considered in the vertical direction: the base water layer and the surface layer. For distributed system models, information on
the geological and topographical characteristics of a river catchment is required to derive or measure the necessary parameters. The river basin characteristics were described by the set of data (elevation, flow direction, catchment area and stream network) derived from the digital elevation model. More details about the performance and validation of the model are presented in Phrakonkham (2019).

### 3.3 Land use change hazards

The scenario in which reduced forest and increased cropland areas are included was first used to assess the impacts of various land use scenarios on the flood hazard map in the present study area. To investigate the sensitive areas of the flood hazard map, this selection was chosen. Hence, the reduction in forest and all forest areas was considered and converted to the worst scenario and to cropland, respectively. One of the suitable geoenvironmental factors of crop fields is slope (Ceballos-Silva and López-Blanco, 2003; Huynh, 2008). As shown by these studies, a slope of approximately 6-12% will increase the
growth of vegetation. Consequently, in the scenario designed first, the forest areas with slope angles less than and more than 12% were converted to cropland and remained unchanged. Second, based on the probability of increased population, an expansion of urban areas was created whose process was represented as moving from rural areas to urban areas.


### 3.4 Landslide hazard

Landslides are one of the most dangerous natural hazards, and they cause major damage to affected areas. To identify the locations of landslide hazard areas throughout Laos, a probabilistic model based on multiple logistic regression analysis was used. The model considers several important physical parameters, including hydraulic and geographical parameters. Among these, the hydrological parameter (i.e., hydraulic gradient) is the most important factor for determining the probability of a landslide (Kawagoe et al., 2010). The statistical approaches used for evaluation are indirect hazard mapping methodologies
that involve a statistical determination based on a combination of variables that have identified land use occurrence (Ohlmacher and Davis, 2003; van Westen et al., 2006). In addition, probabilistic methods are used to determine the probability over a large area where numerous natural slopes exist. Hence, the hydraulic gradient is the main hydraulic parameter. Due to the lack of data in Laos, data from Thailand were used for this study on Laos (Kawagoe et al., 2010; Komori et al., 2018; Ono et al., 2011). In these studies, the probability of a landslide is derived as:


$$p = \frac{1}{1 + \exp[-(-17.494 + 1179.25 \times hydro \times 0.0097 \times relief)]} \tag{1}$$

where $p$ is the probability, which is considered the hazard index of a landslide map, and $hydro$ and $relief$ are the hydraulic gradient and the relative relief, respectively.

Relative relief is defined as the elevation difference between the highest location and lowest location. Relief energy is an
index that can show the complexity of geographical features considering the active development of landforms. Therefore, in this study, relief energy is defined as the elevation difference between the highest and the lowest elevation in each grid cell, and the relief energy for each 1 km×1 km resolution grid cell is estimated using digital elevation model (DEM) data.

Hydraulic gradient is a significant factor for the initiation of landslides. Changes in the hydraulic gradient in the slope area can lead to landslides. In this study, we use unsaturated infiltration analysis based on the Richards equation to find the
change in hydraulic gradient (Kawagoe et al., 2010).

### 3.5 Climate change hazards

Climate change hazards are estimated as a future projection of climate change impacts on future floods and future landslide hazards. The prediction is obtained by the future projection of precipitation from the GCM dataset. In this study, the average precipitation from 7 GCMs (Table s1 from the supplemental material) and three RCP scenarios were selected. Because most
GCMs offer information at scales greater than a few hundred kilometers, statistical downscale bias correction quantile mapping was deployed (Equation (2)) to reduce the bias for precipitation output from the GCMs (Boé et al., 2007; Fajar Januriyadi et al., 2018; Fang et al., 2015; Lafon et al., 2013; Salem et al., 2018). First, the method for bias correction quantile mapping presented by Salem (2018) is used. Then, the near and far future trends in rainfall are chosen as the average future




precipitation data of the GCMs from 2010 to 2050 (2050s) and 2051 to 2099 (2100s). Additionally, the log-Pearson type III

method was used to calculate the return period rainfall for all future rainfall patterns.

$$z_{cor} = CDF_0^{-1}\left(CDF_{gcm}\left(z_{gcm}\right)\right) \tag{2}$$

where $z_{cor}$ is the precipitation after correcting the bias, $z_{gcm}$ is the precipitation from GCMs before bias correction, $CDF_{gcm}$

is the cumulative distribution function (CDF) of $z_{gcm}$ and $CDF_0^{-1}$ is the inverse CDF of the observed rainfall.

### 3.6 Hazard index

### 3.6.1 Flood hazard index classification

We propose a hazard index, which is adapted from the relationship between velocity and flood depth (Sally et al., 2008). By

considering the water depth of every grid in the flood map, we converted the value to a hazard index. The scenario was as

follows: the water velocity from the flooded areas was low, and the depth can be transformed into a hazard index. The index

is scaled from zero to one, with zero representing the lowest hazard and one representing the highest hazard. The hazard

index was classified into four categories, i.e., small, medium, high and very high hazards, which correspond to inundation

depths of 0.0-0.3, 0.31-0.6, 0.61-2.0 and more than 2.1 m, respectively. Subsequently, we can find the relationship between

flood depth and hazard index, as shown in Figure 1, and the flood depth and hazard index curve can be derived

### 3.6.2 Landslide hazard index classification

The probability of landslides (0-1) is used directly as the landslide hazard index (0-1). The landslide hazard map was

classified using the natural breaks method provided in the ArcGIS program. The natural breaks method is a data

classification method designed for determining the best arrangement in terms of representing the spatial distribution of the

data (Bednarik et al., 2010; Constantin et al., 2011; Erener and Düzgün, 2010; Falaschi et al., 2009; MohanV and RajT,

2011; Pourghasemi et al., 2012). In this study, we want to classify our data into 4 classes that are similar to flood hazard

maps for convenience and for comparison to other hazard maps. Finally, the landslide hazard map is graded into 4 classes:

low (0-0.23), medium (0.23-0.54), intermediate (0.54-0.85) and high (0.85-1).

### 3.7 Analytical Hierarchy Process (AHP)

The AHP method is a highly efficient method among multicriteria decision-making approaches. This method can prioritize

multicriterion data using a pair comparison approach (Saaty, 1994). In a previous study (Phrakonkham et al., 2019), we

conducted a questionnaire survey with expert officers overseeing various hazards and risks in Laos. In the survey

questionnaires, experts were asked to provide their judgments on three hazards: floods, land use changes and climate change

impacts on floods. In the present study, however, five hazards are asked in the questionnaires. We have 5 criteria, which

include floods, land use changes, landslides, climate change impacts on floods and climate change impacts on landslides;





thus, the matrix is 5 by 5, and the diagonal elements are equal to 1. The value of each row of pairwise comparisons is determined based on expert judgments.

To obtain the criteria relative priority value, expert judgments are required. We designed and conducted a questionnaire at the Ministry of Natural Resource and Environment of Laos because most of the officers who work in this ministry have knowledge of flood hazards, climate changes, and land use impacts in Laos (Table s2). All the experts and those who had experience in the field of our concerned hazards were asked to complete a questionnaire. Approximately 41 samples were collected from all the expert officers at the Ministry of Natural Resource and Environment. By using Equation (3), we
obtained a value for each pairwise comparison from each row of the questionnaires.

$$Rel_j = \sqrt[m]{\frac{\prod_{i=1}^{m} A_{m,j}}{\prod_{i=1}^{m} B_{m,j}}} \tag{3}$$

where $Rel_j$ is the relative importance of the pairwise criteria in the $j$th row from the questionnaire; for example, row $j = 1^{st}$
represents the pairwise comparison between flood and land use change, and $m$ is the number of samples (in this study, $m = 41$).

According to Saaty (1994), the weight ($w_i$) is the normalized eigenvector of the matrix ($D_{i,k}$) associated with the largest eigenvalue $\lambda_{max}$ of the matrix ($D_{i,k}$). $w_i$ ($i = 1, 2,…, 5$) is the weight of each hazard corresponding to the hazard from the $ith$ row of Table 1; for example, $w_1$ ($i = 1$) is the weight of the flood hazard ($w_1 = w_{flood}$) according to Table 1 ($w_2 =$
$w_{land\ use\ change}$, $w_3 = w_{landslide}$, $w_4 = w_{climate\ change\ to\ flood}$ & $w_5 = w_{climate\ change\ to\ landslide}$). The weights for the pairwise comparison matrix are presented in Table 1. After we obtain the weights of each hazard, its consistency must be evaluated if the consistency ratio is less than 0.1. More details about consistency can be found in Saaty (1994). In this study, the calculated consistency ratio was 0.03, indicating that the results from the questionnaire were consistent.

**3.8 AHP-based hazard map**

To integrate the above flooding, land use, landslides, climate change leading to floods and climate change leading to landslides hazard maps, the AHP-based hazard index is used. This index is also deployed to assimilate the weight of each criterion used to assign its role in the final map. Each grid must therefore be evaluated based on all criteria. The AHP-based hazard index can be derived as follows:

$AHP_{\bar{x},\bar{z}} hazard\ index = \left(HI_{\bar{x},\bar{z},flood} \times w_{flood}\right) + \left(HI_{\bar{x},\bar{z},land\ use\ change} \times w_{land\ use\ change}\right) + \left(HI_{\bar{x},\bar{z},land\ slide} \times$

$w_{land\ slide}\right) + \left(HI_{\bar{x},\bar{z},climate\ change\ to\ flood} \times w_{climate\ chnage\ to\ flood}\right) + \left(HI_{\bar{x},\bar{z},climate\ change\ to\ landslide} \times$

$w_{climate\ chnage\ to\ landslide}\right) \tag{4}$





where $HI_{\bar{x},\bar{z},flood}$ ( $\bar{x}$ = 1, 2,….., $\overline{xx}$ ; $\bar{z} = 1, 2, ….., \overline{zz}$ ) is a hazard index value from the flood hazard map;
$HI_{\bar{x},\bar{z},land\ use\ change}$, $HI_{\bar{x},\bar{z},land\ slide}$, $HI_{\bar{x},\bar{z},climate\ change\ to\ flood}$, and $HI_{\bar{x},\bar{z},climate\ change\ to\ landslide}$ are hazard index values
from land use change, landslides, climate change leading to floods and climate change leading to landslides hazard maps,
respectively; $\bar{x}$ is a vertical coordination grid on the map; and $\bar{z}$ is a horizontal coordination grid on the map. Every hazard
map (flood, landslide, and so on) has an equal number of horizontal and vertical grids; $\overline{xx}$ is the number of vertical grids, and
$\overline{zz}$ is the number of horizontal grids from the hazard map. For the classification of integrated hazard maps, we apply the
natural break method from section 4.6.2 for the classification because the method can determine the best arrangement of
values into different classes. The integrated hazard map was classified into four hazard areas corresponding to low (0-0.21),
medium (0.22-0.43), high (0.44-0.68) and very high hazard (0.69-1.0) areas.

## 4 RESULTS

### 4.1 Flood hazard map

A distributed hydrological model was used to simulate a flood hazard map for the whole country. We considered the greatest
water depth in every grid cell, which was determined by contributing factors during the simulation, and these factors
included the 100-year return periods of rainfall, land types, soil hydrologic characteristics, and elevation. The results are
shown in Figure 2, where we can see the potential flood hazard areas. The results reveal that low-hazard areas cover 78.44%
of the total area, medium-hazard areas cover 12.64%, and high- and very high-hazard areas cover 6.14% and 2.78%,
respectively.

### 4.2 Landslide hazard map

According to the results shown in Figure 3, most of the hazard areas are located around the central to southern parts of Laos.
In addition, the records of landslide events in Laos show that those landslide events are closely related to the probability of
exceeding values of rainfall. The results reveal that the low-hazard areas cover 92.67%, the medium-hazard areas cover
1.83%, the high-hazard areas cover 1.21% and the very high-hazard areas cover 4.28% of the total area. The landslide hazard
map was validated by comparing the landslide hazard map results with historical landslide events in Lao PDR, in which
those events occurred with the extreme rainfall of a 100-year return period. Approximately 33 landslide events (Figure 3)
were used for comparison with the landslide hazard map results. From the results, 22 events (66.67%) were located in very
high-hazard areas, 8 events (24.24%) were located in high-hazard areas, and 3 events (9.09%) were located in low-hazard
areas. The landslide hazard map by our simulation corresponds to the historical landslide events in the country. These results
confirm that the landslide model and landslide hazard map can predict the occurrence of landslides in Lao PDR.





### 4.3 Land use change hazard map

The results in Figure 4 show the overall impact of the hazard areas, which are growing significantly; this is mostly because of the loss of forest area that slows the rainfall runoff. Without forest area, all rainfall runoff runs directly downstream without storage or other factors to slow it down. Therefore, the hazard areas downstream are expanding. The total area of land use change impacts on floods was divided into 77.08%, 12.68%, 6.94% and 3.3% of low-, medium-, high- and very high-hazard areas, respectively.

### 4.4 Climate change hazard map

### 4.4.1 Climate change impact on floods hazard map

Developing countries in tropical regions are highly susceptible to floods. These regions already have high levels of precipitation, and the hydrologic cycle is significantly interlinked and sensitive to the weather. Future scenarios of flood hazard maps for the near and far future under three scenarios are shown in Figure 5. The percentage of very high-hazard
areas for the near future increased from 3.71% under RCP2.6 to 4.05% under the RCP8.5 scenario; additionally, for the far future, the percentage of very high-hazard areas increased from 4% under the RCP2.6 scenario to 4.88% under RCP8.5. In the climate change hazard map with respect to the change in the flood hazard map, under all scenarios, the maximum high-hazard areas were 0.33% in urban areas, 88.77% in forest areas, 2% in paddy field areas and 9.0% in agricultural areas. It was also seen that the very high-hazard areas represented 0.35, 90.09, 1.8 and 7.77% of the urban, forest, paddy field and
agricultural areas, respectively.

### 4.4.2 Climate change impact on landslides hazard map

Future landslides under the three scenarios and two time periods were simulated (Figure 6). The percentage of very high-hazard areas for the near future increased from 3.71% under RCP2.6 to 4.05% under the RCP8.5 scenario; additionally, for the far future, the percentage of very high-hazard areas increased from 4% under the RCP2.6 scenario to 4.88% under
RCP8.5. In the climate change hazard map with respect to the change in the landslide hazard map, under all scenarios, the maximum high-hazard areas were 0.13% in urban areas, 88.98% in forest areas, 0.84% in paddy field areas and 10.05% in agricultural areas. It was also seen that the very high-hazard areas represented 0.15, 90.31, 0.77 and 8.77% of the urban, forest, paddy field and agricultural areas, respectively.

### 4.5 Integrated hazard maps

The main objective of this chapter is to integrate the five existing hazard maps (floods, landslides, land use changes, climate change impacts on floods and climate change impacts on landslides). Phrakonkham (2019) proposed the AHP-based method for integrated multihazard maps in Lao PDR, namely, flood, land use change and climate change leading to flood hazard maps. Based on the results, the AHP-based integrated hazard map can show potential hazard areas at the country scale. In



this study, 6 integrated hazard maps under the 3 RCP scenarios (RCP2.6, RCP4.5 and RCP8.5) and the 2 time periods (near-
future (2050s) and far-future (2100s)) were produced using the AHP method (Figure 7). The integrated hazard maps were
categorized using the natural breaks method of classification (Tate et al., 2010). It was noticeable that the total amount of
very high-hazard areas increased in response to the RCP scenarios. In the near future, the percentage of very high-hazard
areas increased from 3.20% under RCP2.6 to 3.3% under RCP8.5. Similar results are shown for the far future; the proportion
of high-hazard areas increases from 3.23% under RCP2.6 to 3.71% under RCP8.5.

To validate the performance of the integrated hazard maps, 30 historical flood events and 33 historical landslide events were
compared to the integrated hazard maps (Figure 8). According to the results, for historical flood events, 2 events (7%) were
located in low-hazard areas, 3 events (10%) were located in medium-hazard areas, 14 (46%) events were located in high-
hazard areas and 11 (37%) events were located in very high-hazard areas. For historical landslide events, 7 (21%) events
were located in low-hazard areas, 8 (24%) events were located in medium-hazard areas, 11 (33%) events were located in
high-hazard areas and 7 (21%) events were located in very high-hazard areas. The majority of historical landslide (54%) and
flood (83%) events were located in high- and very high-hazard areas. Hence, the reliability of the integrated hazard map was
confirmed.

## 5. Discussion

Flood hazard maps have demonstrated the distribution of hazard areas across the study area. Notably, most of the hazard area
distributions were located in the central and southern regions of Lao PDR. Vientiane is located in the central region, and
little of the area in the Vientiane capital area is impacted by flood hazards. Based on the results, a high-hazard area is visible
around the central-southern region of Lao PDR. High- and very high-hazard areas in each province were divided by the
whole country area to obtain their proportions of hazard areas (Table s3). For very high-hazard areas, Bolihamxai (0.27%),
Savannakhet (0.27%) and Vientiane Provinces (0.26%) have the highest percentage of very high-hazard areas. For the
capital of Lao PDR, only 0.08% of total high-hazard areas and 0.04% of total very high-hazard areas are located in
Vientiane, and the city has the lowest percentage of total high- and very high-hazard areas among all the provinces.
Champasak is one of the large provinces and developed areas of Lao PDR. Approximately 0.45% of the total high-hazard
area and 0.18% of the total very high-hazard area are located in Champasak Province. Compared to Vientiane Capital,
Champasak has higher proportions of both high- and very high-hazard areas.

The landslide hazard map shows the distribution of potential hazard areas from landslides around mountains in the central
and southern regions. According to the results, most of the landslide hazard areas are located in forest areas, followed by
agricultural areas and paddy fields. Most agricultural and paddy field areas belong to ethnic groups that have livelihoods
near mountainous areas. In Lao PDR, for many ethnic groups living in mountainous areas, their sources of income are
mainly from agricultural production. Compared to other provinces of Lao PDR, Xiangkoung, Blolikhamxai and Vientiane
have high mountainous areas; for instance, Bolikhamxai has the highest percentage of high-hazard areas (0.48%) (Table s4).




For very high-hazard areas, Bolikhamxai Province has the highest percentage areas (2.31%). Based on historical landslide events from Figure 3, Xiangkoung, Bolikhamxai and Vientiane are three provinces in which several landslides occurred. Xiengkoung has approximately 0.6% of very high-hazard areas, Bolikhamxai has approximately 2.31% and Vientiane has 0.92% of very high-hazard areas. These provinces should be given priority for developing mitigation and countermeasures.

Most of the mountainous areas in these provinces provide livelihoods for different ethnic groups. Therefore, most landslide hazards occurring in these areas will have a direct impact on agriculture and the properties of ethnic groups.

The land use change hazard map shows a distribution similar to that of the flood hazard map but with a higher magnitude. Overall, the high-hazard areas and very high-hazard areas increase when comparing the land use change hazard map to the flood hazard map (Table s5 and Table s6). The high-hazard areas of the land use change hazard map increase by

approximately 13%, and the very high-hazard areas increase by approximately 19% compared to the high- and very high-hazard areas of the current flood hazard map. Similar to the flood hazard map, Savannakhet Province has the highest percentage of high- (0.96%) and very high-hazard areas (0.3%). However, compared to the flood hazard map, the high- and very high-hazard areas of Savannakhet Province slightly increased. The Vientiane Capital area had a greater impact than that of Champasak Province. The very high-hazard area in Vientiane Capital increases by approximately 82%, and the high-

hazard area increases by 60%. It is indicated that Vientiane Capital is more highly influenced than Champasak Province by land use change. It is indicated that land use change has a significant influence on the magnitude of flooding area. The results correspond to Huntington (2006), who found that land use change from human alterations such as the conversion of forest area to agricultural area or the expansion of urban area will lead to an increase in flood hazard area.

Climate change impacts on flood hazard maps are represented by the flood hazard map under future climate conditions with

3 scenarios (RCP2.6, 4.5 and 8.5) and 2 time periods (near future and far future). The flood hazard area under the influence of future rainfall conditions shows an increase across the country. By considering the near future period (Figure s3), for instance, Luang Namtha Province has the highest increase (23%) of very high-hazard areas when comparing the flood hazard map under scenario RCP2.6 to that under RCP4.5 (Table s7). In Bolikhamxai Province, the highest increase (5%) of very high-hazard areas was observed when comparing the flood hazard maps under scenarios RCP4.5 and RCP8.5 (Table s8). For

the far future period, the total percentage of very high-hazard area increases from 4% under the RCP2.6 scenario to 4.22% under the RCP4.5 scenario, and it increases to 4.88% under the RCP8.5 scenario (Figure s4). In many provinces, the climate change impacts on flood hazard maps in the near and far future have continuously increasing very high-hazard areas from RCP2.6 to RCP8.5 (Tables s9 and s10). In addition, the future rainfall projections under the RCP2.6, 4.5 and 8.5 scenarios match the increases in the very high flood-hazard areas under the RCP4.5 to RCP2.6 and RCP8.5 to RCP4.5 scenarios

(Figure s5 and s6). Overall, the amount of rainfall increases, particularly in Khammouan, Bolikhamxai and Attapeu Provinces, which is in line with the results.

Climate change impacts on the landslide hazard map are represented by the landslide hazard map under future climate conditions with 3 scenarios and 2 time periods. By considering the near-future period (Figure s7), the total percentage of very high-hazard area of 4.85% under the RCP2.6 scenario increases to 4.92% under the RCP4.5 scenario and increases to





4.96% under the RCP8.5 scenario. The climate change impacts on landslide hazard maps in the near future in many
provinces have continuously increasing very high-hazard areas from RCP2.6 to RCP8.5 (Table s11 and Table s12). For the
far future (Figure s8), comparing the increase in the very high-hazard area between future landslides under the RCP2.6 and
RCP4.5 scenarios, Bolikhamxai Province has the highest increase from 2.93% under the RCP2.6 scenario to 3.2% under the
RCP4.5 scenario (Table s13). Bolikhamxai Province has the highest increase (5%) in the very high-hazard area when

comparing the landslide hazard maps under the RCP4.5 and RCP8.5 scenarios (Table s14). In most of the provinces, the very
high-hazard area from climate change impacts on landslide hazard maps increases continually in the far future from RCP2.6
to RCP8.5, for example, in Bolikhamxai Province. Based on the results, the increase in rainfall intensity (Figure s5 and
Figure s6) due to climate change influences the increase in flood and landslide hazard areas. Many studies in the Mekong
Delta (Dinh et al., 2012; Lauri et al., 2012) revealed that climate change has impacts on rainfall intensity, which leads to

increases in flood and landslide frequencies. Therefore, these results are in line with those of other research studies.

The integrated maps consist of flooding, land use change, landslide and climate change hazards. The maps are developed
using the AHP to perform the integration. The integrated hazard map consists of 6 maps under 3 RCP scenarios and 2 time
periods. Figure 9 (d) shows the area of the hazard index increase when comparing the integrated hazard map for the near
future under the RCP2.6 and RCP4.5 scenarios. Savannakhet Province is highly influenced by climate change. The

percentage of the very high-hazard area from the integrated hazard map increases by approximately 4.69% when comparing
the RCP2.6 and RCP4.5 scenarios (Table 2). Figure 9 (e) shows the area of the hazard index increase when comparing the
RCP8.5 and RCP4.5 scenarios. Among others, Khammouan, Vientiane, Savannakhet and Bolikhamxai Provinces have
higher increases in very high-hazard areas when comparing integrated hazard maps under the RCP4.5 and RCP8.5 scenarios
(Table 3). For the far future period, Figure 10 (d) shows the area of the hazard index increase when comparing the RCP4.5

and RCP2.6 scenarios. Comparing the increase in the very high-hazard area between the integrated hazard map under the
RCP 2.6 and RCP 4.5 scenarios, Khammouan Province has the highest increase (16.45%) (Table 4). Figure 10 (e) shows the
area of the hazard index increase when comparing the RCP8.5 and RCP4.5 scenarios. Khammouan Province has the highest
increase in the very high-hazard area (12.47%) when comparing the flood hazard maps under the RCP2.6 and RCP4.5
scenarios (Table 5). The increase in the very high-hazard areas for the integrated hazard map is similar to that for the rainfall

patterns from the RCP2.6 to RCP4.5 and RCP4.5 to RCP8.5 scenarios with near- and far-future periods (Figure s5 and
Figure s6). The southern region has the highest increase in very high-hazard areas, particularly Bolikhamxai, Khamouan and
Savannakhet Provinces. Special attention must be paid to these provinces, particularly to countermeasures and adaptation
planning, to reduce the potential risk. The produced integrated hazard map identified suitable areas for development in the
northern part of Laos, which had the greatest amount of low-hazard areas (42%).

The existing studies on multihazard mapping mainly focus on aggregating all individual hazards with equal weight, the sum
of the hazard indexes from individual hazards or using the frequency of occurrence for each hazard to decide the weight,
which does not sufficiently reflect the various impacts of different hazards present in the same area. In addition, those studies
have not considered the participation of stakeholders. New concepts in this study are that we take into account the opinions



of stakeholders by comparing each individual hazard to find the importance of each hazard. The importance of each
individual hazard was determined by the AHP method. Furthermore, AHP is a method that attempts to imitate human
rationality for decision making by using experiences and perceptions from stakeholders and experts. It offers the
organization of knowledge, simplifies structures for understanding the issue and consistency, and involves human logic and
intuition as well as experiences. In addition, the pairwise comparisons help stakeholders and experts focus their judgment on
each comparison criterion. Each criterion has a certain value that represents a judgment of the likelihood of its scale of
importance to that of others. The integrated hazard map based on AHP can identify the potential distribution of hazard areas
across the country. In addition, the integrated map can provide the preliminary results for the distribution pattern of hazard
areas; furthermore, the damage cost from the potential risk area can be estimated. Moreover, the integrated hazard map can
be used in combination with other maps, such as the future development plan from the government or private sectors. In this
way, the areas of hazard in the development of agricultural areas or the expansion of urban areas could be verified. These
maps are applicable to the presentation of the spatial distribution of hazard areas.

## 6. Conclusions

The main objective of this study was to develop an integrated hazard map that is reliable at the national scale. The integrated
maps apply the AHP method for integrating all individual hazard maps together, namely, flooding, land use change,
landslides and climate change impacts on flood hazards and climate change impacts on landslide hazard maps. This study
provides a significant and valid methodology for the development of integrated hazard maps using multicriteria decision
analysis, such as AHP. The results from integrated hazard maps can identify dangerous areas from both individual and
integrated hazards. In addition, the results can be used as primary data for screening and selecting development areas.
However, it should be noted that data on population and economic impacts in hazard areas are not yet included in this study.
Together with population and economic data in hazard areas, risk areas could be identified.

Data availability:The data used in this paper are available from the authors upon request.

Author contributions: PS contributed to the literature review, methodology, original draft preparation, major and minor
revisions, and proofreading. KS contributed to the conceptualization, methodology, revisions, and proofreading. SW
contributed to revisions and proofreading.

Competing interests: The authors declare that they have no conflict of interest.

Acknowledgements: This research was partially supported by the Grants-in-Aid for Scientific Research (B), 2015-2017
(15H05218, So Kazama) from the Ministry of Education, Science, Sports and Culture. The authors thank the Environmental
Research and Technology Development Fund (s-14) from the Ministry of the Environment, Japan. Special thanks are
extended to Dr. Sengprasong Phrakonkham for his support and assistance in conducting questionnaires for the AHP analysis

Review statement.



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




| Velocity (m/s) | Depth of flooding (m) | | | | | | | | | | | |
|---|---|---|---|---|---|---|---|---|---|---|---|---|
| | 0.05 | 0.1 | 0.2 | 0.3 | 0.4 | 0.5 | 0.6 | 0.8 | 1 | 1.5 | 2 | 2.5 |
| 0 | | | | | | | | | | | | |
| 0.1 | | | | | | | | | | | | |
| 0.25 | | | | | | | | | | | | |
| 0.5 | | | | | | | | | | | | |
| 1 | | | | | | | | | | | | |
| 1.5 | | | | | | | | | | | | |
| 2 | | | | | | | | | | | | |
| 2.5 | | | | | | | | | | | | |
| 3 | | | | | | | | | | | | |
| 3.5 | | | | | | | | | | | | |
| 4 | | | | | | | | | | | | |
| 4.5 | | | | | | | | | | | | |
| 5 | | | | | | | | | | | | |

| Flood depth (m) | Hazard index |
|---|---|
| Small hazard < 0.3 | 0-0.25 |
| Medium hazard < 0.6 | 0.25-0.5 |
| High hazard< 2 | 0.5-0.75 |
| Very high hazard > 2 | 0.75-1 |

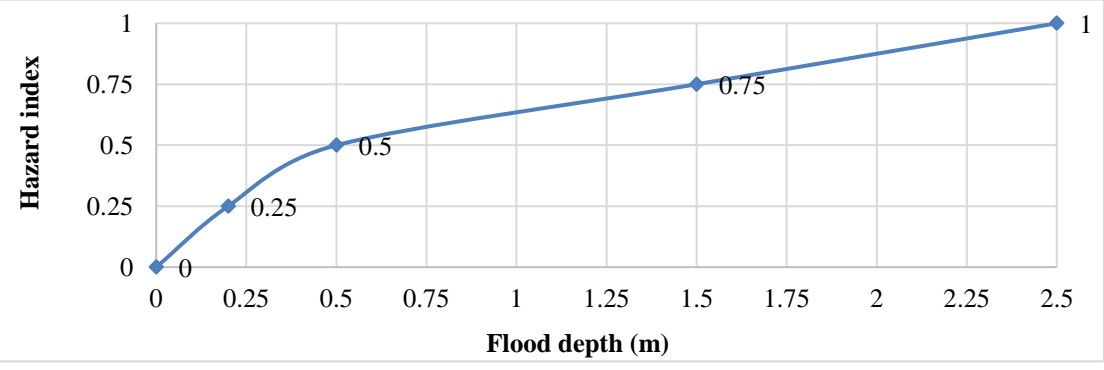

**Figure 1. Flood depth-velocity relationship to the hazard index and curve.**


**Figure 2. Flood hazard map.**



**Figure 3. Landslide hazard map and historical landslide events.**






**Figure 4. Land use change hazard map.**


2050s          2100s



**Figure 5. Flood hazard maps with the ensemble average of heavy rainfall from the 7 GCMs that used data under the RCP2.6, RCP4.5 and RCP8.5 scenarios.**



**2050s**   **2100s**

**RCP 2.6**

**RCP 4.5**

**Hazard index**

- Low hazard
- Medium hazard
- High hazard
- Very high hazard

**RCP 8.5**

**Figure 6. Landslide hazard maps with the ensemble average of heavy rainfall from the 7 GCMs that used data under the RCP2.6, RCP4.5 and RCP8.5 scenarios.**


**Figure 7. Integrated hazard maps with the ensemble average of heavy rainfall from the 7 GCMs that used data under the RCP2.6, RCP4.5 and RCP8.5 scenarios.**



**Figure 8. Comparison of historical flood and landslide events to the integrated hazard map of scenario RCP2.6 during the near future.**
(a)

(b)

(c)

(d)

(e)

**Figure 9. Integrated hazard maps for the 100-year return period under scenarios (a) RCP2.6, (b) RCP4.5, and (c) RCP8.5 and the difference in hazard index between scenarios (d) RCP4.5 and RCP2.6 and between scenarios (e) RCP8.5 and RCP4.5 during the near future.**




**Figure 10. Integrated hazard maps for the 100-year return period under scenarios (a) RCP2.6, (b) RCP4.5, and (c) RCP8.5 and the difference in hazard index between scenarios (d) RCP4.5 and RCP2.6, and between scenarios (e) RCP8.5 and RCP4.5 during the far future.**




**Table 1. AHP pairwise comparison matrix ($D_{i,k}$) with the weight of each criterion.**

| Option B ($k$) \ Option A ($i$) | $k=1$ Flood | $k=2$ Land use change | Landslide | $k=5$ Climate change leading to floods | Climate change leading to landslides | Weight ($w_i$) |
|---|---|---|---|---|---|---|
| Flood | 1.00 | 4.20 | 7.10 | 0.71 | 4.10 | $i=1$ |
| Land use change | 0.24 | 1.00 | 3.60 | 0.18 | 1.60 | $i=2$ |
| Landslide | 0.14 | 0.28 | 1.00 | 0.17 | 0.34 | |
| Climate change leading to floods | 1.4 | 5.4 | 5.7 | 1.00 | 5.50 | 0.42 |
| Climate change leading to landslides | 0.24 | 0.63 | 2.9 | 0.18 | 1.00 | 0.09 |
| Sum | 3.02 | 11.50 | 20.30 | 2.26 | 12.54 | $i=5$ |








**Table 2. Percentage of very high-hazard area from the integrated hazard map in each province and the percentage of increase between the RCP4.5 and RCP2.6 scenarios during the near future.**

| Province name | Percentage of very high-hazard area under RCP2.6 | Percentage of very high-hazard area under RCP4.5 | Percentage increase in very high-hazard area between RCP4.5 and 2.6 |
|---|---|---|---|
| Attapeu | 0.23% | 0.23% | 0.31% |
| Bokeo | 0.07% | 0.07% | 0.64% |
| Bolikhamxai | 0.32% | 0.33% | 3.05% |
| Champasak | 0.21% | 0.22% | 0.28% |
| Houaphan | 0.22% | 0.22% | 0.20% |
| Khammouan | 0.32% | 0.32% | 0.94% |
| Louang Namtha | 0.08% | 0.08% | 4.36% |
| Louang Prabang | 0.19% | 0.20% | 4.21% |
| Oudomxai | 0.12% | 0.12% | 3.47% |
| Phongsaly | 0.11% | 0.11% | 1.03% |
| Salavan | 0.13% | 0.13% | 1.18% |
| Savannakhet | 0.36% | 0.38% | 4.69% |
| Vientiane | 0.30% | 0.31% | 2.86% |
| Vientiane Capital City | 0.04% | 0.04% | 0.34% |
| Xaignabouly | 0.19% | 0.20% | 1.80% |
| Xekong | 0.14% | 0.14% | 1.30% |
| Xiangkouang | 0.17% | 0.17% | 1.56% |
| Total percentage of very high-hazard area across the country | 3.2% | 3.27% | |




**Table 3. Percentage of very high-hazard area from the integrated hazard map in each province and the percentage of increase between the RCP8.5 and RCP4.5 scenarios during the near future.**

| Province name | Percentage of very high-hazard area under RCP4.5 | Percentage of very high-hazard area under RCP8.5 | Percentage increase in very high-hazard area between RCP8.5 and 4.5 |
|---|---|---|---|
| Attapeu | 0.23% | 0.23% | 0.98% |
| Bokeo | 0.07% | 0.07% | 0.29% |
| Bolikhamxai | 0.33% | 0.34% | 1.43% |
| Champasak | 0.22% | 0.22% | 0.92% |
| Houaphan | 0.22% | 0.22% | 0.95% |
| Khammouan | 0.32% | 0.32% | 1.37% |
| Louang Namtha | 0.08% | 0.08% | 0.34% |
| Louang Prabang | 0.20% | 0.20% | 0.87% |
| Oudomxai | 0.12% | 0.12% | 0.52% |
| Phongsaly | 0.11% | 0.11% | 0.48% |
| Salavan | 0.13% | 0.13% | 0.54% |
| Savannakhet | 0.38% | 0.39% | 1.62% |
| Vientiane | 0.31% | 0.32% | 1.34% |
| Vientiane Capital City | 0.04% | 0.04% | 0.16% |
| Xaignabouly | 0.20% | 0.20% | 0.84% |
| Xekong | 0.14% | 0.14% | 0.60% |
| Xiangkouang | 0.17% | 0.17% | 0.72% |
| Total percentage of very high-hazard area across the country | 3.27% | 3.3% | |




**Table 4. Percentage of very high-hazard area from the integrated hazard map in each province and the percentage of increase between the RCP4.5 and RCP2.6 scenarios during the far future.**


| Province name | Percentage of very high-hazard area under RCP2.6 | Percentage of very high-hazard area under RCP4.5 | Percentage increase in very high-hazard area between RCP4.5 and 2.6 |
|---|---|---|---|
| Attapeu | 0.23% | 0.25% | 8.67% |
| Bokeo | 0.07% | 0.07% | 2.58% |
| Bolikhamxai | 0.33% | 0.37% | 12.39% |
| Champasak | 0.22% | 0.23% | 8.16% |
| Houaphan | 0.22% | 0.24% | 8.44% |
| Khammouan | 0.32% | 0.37% | 16.45% |
| Louang Namtha | 0.08% | 0.08% | 2.90% |
| Louang Prabang | 0.20% | 0.21% | 7.41% |
| Oudomxai | 0.12% | 0.12% | 4.48% |
| Phongsaly | 0.11% | 0.12% | 4.17% |
| Salavan | 0.13% | 0.13% | 4.77% |
| Savannakhet | 0.37% | 0.41% | 11.35% |
| Vientiane | 0.31% | 0.34% | 11.62% |
| Vientiane Capital City | 0.04% | 0.04% | 1.37% |
| Xaignabouly | 0.19% | 0.21% | 7.28% |
| Xekong | 0.14% | 0.15% | 5.26% |
| Xiangkouang | 0.17% | 0.18% | 6.31% |
| Total percentage of very high-hazard area across the country | 3.23% | 3.52% | |



**Table 5. Percentage of very high-hazard area from the integrated hazard map in each province and the percentage of increase between the RCP8.5 and RCP4.5 scenarios during the far future.**

| Province name | Percentage of very high-hazard area under RCP4.5 | Percentage of very high-hazard area under RCP8.5 | Percentage increase in very high-hazard area between RCP8.5 and 4.5 |
|---|---|---|---|
| Attapeu | 0.25% | 0.25% | 1.36% |
| Bokeo | 0.07% | 0.07% | 1.42% |
| Bolikhamxai | 0.37% | 0.41% | 11.90% |
| Champasak | 0.23% | 0.24% | 2.77% |
| Houaphan | 0.24% | 0.25% | 3.78% |
| Khammouan | 0.36% | 0.41% | 12.47% |
| Louang Namtha | 0.08% | 0.08% | 1.60% |
| Louang Prabang | 0.21% | 0.21% | 0.99% |
| Oudomxai | 0.12% | 0.13% | 0.66% |
| Phongsaly | 0.12% | 0.12% | 1.13% |
| Salavan | 0.13% | 0.13% | 0.59% |
| Savannakhet | 0.42% | 0.46% | 10.72% |
| Vientiane | 0.34% | 0.37% | 8.33% |
| Vientiane Capital City | 0.04% | 0.04% | 0.75% |
| Xaignabouly | 0.21% | 0.21% | 0.62% |
| Xekong | 0.15% | 0.15% | 0.77% |
| Xiangkouang | 0.18% | 0.18% | 1.00% |
| Total percentage of very high-hazard area across the country | 3.52% | 3.71% | |
