# Peer review of "Integrated evaluation mapping of water-related disasters using the analytical"

_Natural Hazards and Earth System Sciences, 2020_

## Referee Comment (RC1) · Anonymous Referee #1 · 15 Sep 2020

Generally, this is an important work for a data sparse country. Even though authors tried to develop multi-hazard maps, there are issues with this work. Hence a round of revision is essential. I have outlined my comments below to consider for improvement: [1] Line 19: a comma is essential after country [2] line 22: instead of 'can lead to' you may change to 'can increase' [3] there are a number of important works in this space which require attention. It seems that the current version lacks of international significance of this work. Hence think they may consider the following works to improve its readership. Furthermore, authors reviewed existing works but missed

many in the area https://royalsocietypublishing.org/doi/pdf/10.1098/rsos.191957 https://www.sciencedirect.com/science/article/pii/S2212420920312632 https://www.sciencedirect.com/science/article/pii/S0264837720305470 https://www.nature.com/articles/s41598-020-69233-2 Line 66: There is a big contraction to your aim described here with that of aim in the abstract section. In the abstract, you are stating to develop an approach but here is a reliable flood hazard map? Which one is correct? This requires serious attention Section 3.2: What do you mean by expressions in lines 108-109? Unclear. What was the resolution of DEM and what was the vertical accuracy of the model? Clarify Section 3.6.1 This section requires describing the method clearly, how have you done this? Existing texts do not support this Line 174: should be "we wanted to.." Line 185: How they have been chosen? At random? Was there any ethical permission sought? What were the main elements of questionnaire? Discussion section is not properly reflecting what are you trying to achieve relative to your objective(s). Specifically, analyse and interpret your findings with the aid of theory, show similarities, dissimilarities. How your finding(s) differs from theory? Existing works showed above may be of help. Conclusion section is also need improvements. What are the limitations? What are the take-home message(s) of this work? Nothing is clear. As it currently stands, conclusion section is sketchy and does not lead to useful conclusion(s) Reduce number of maps in the work, show only crucial ones and the rest can go into Sup Info

---

## Referee Comment (RC2) · Anonymous Referee #2 · 25 Nov 2020

The authors proposed and reported an integrated mapping approach in the context of the data-poor region which is promising. This can be very relevant within the scope of NHESS.

The empirical evidence is valuable for making climate-friendly development policy for many vulnerable countries with less economic advancements. The authors reported most of the fundamental elements for meeting the international and sound scientific standards; however, it may need to revise further before taking a publication decision. Here are some of the specific observations:

[Figure]

The title seems to belong to and less declarative. Changing to "Mapping" might be a good fit than "Evaluation"

The abstract may be improved – highlighting generalization of results and limitations of this study approach

The introduction may restructure – pushing the facts about the case study (national) a bit later, better say something at the very beginning about international facts as a motivation of this study

It is understandable, the author is introducing the AHP as a method in the introduction; however, the objective comes very late. Here it may help to be short, but specific to the research gap. Anyway, AHP related discussion is also part in the method section.

In the methodology, it remains unclear –about sensitivity analysis. It was done or not! If not why not?

Under land use – only "forest and cropland" has been considered – is it because of data availability?

AHP is a popular method for making an expert judgement; however, it can be very complex and time-consuming to communicate with the expert respondents; it might be interesting for the readers to learn from your experiences. Moreover, what are the criteria for being an expert for answering your AHP Matrix?

Some of the discussion may help – why not other MCA approaches was considered like ANP. . ..

There is a number of literatures that has been already included – it might be relevant to look more on: - https://www.sciencedirect.com/science/article/abs/pii/S2212420915301023 - https://www.researchgate.net/profile/Asad$_{Asadzadeh}$/publication/271065059$_A$ssessing$_s$ite$_s$election$_o$f$_n$ew$_t$owns$_u$sing$_T$O.

The presentation of the results needs to be improved further. For example, the cartographic presentation e.g. colour combination may rethink for better visualization of results. For example, following the presentation of the whole study area map, it will be nice to see some high-resolution map by zoom on some specific critical area for a close look at the output.

The discussion might be highlighted about the combined experience of multiple data sources, what are the major challenges. So far you have been using open data and automated workflow!! How about transferability and reproducibility of your proposed approach for countries that are having similar context and challenges.

The conclusion may summarize the significant results and contributions (i.e. in bullet points).

Please also note the supplement to this comment:
https://nhess.copernicus.org/preprints/nhess-2020-195/nhess-2020-195-RC2-supplement.pdf

**Supplement:**

The authors proposed and reported an integrated mapping approach in the context of data poor region which is promising. This can be a very relevant within the scope of NHESS. The empirical evidences are valuable for making climate friendly development policy for many vulnerable countries with less economic advancements. The authors reported most of the fundamental elements for meeting the international and sound scientific standards; however, it may need to revise further before taking a publication decision. Here are some of the specific observations:

- The title seems to be long and less declarative. Changing to "Mapping" might be a good fit than "Evaluation"
- The abstract may be improved – highlighting generalization of results and limitations of this study approach
- Introduction may restructure – pushing the facts about the case study (national) a bit later, better say something at the very beginning about international facts as a motivation of this study
- It is understandable, the author is introducing the AHP as a method in the introduction; however, the objective comes very late. Here it may help to be short, but specific to the research gap. Anyway, AHP related discussion are also part in the method section.
- In the methodology, it remain unclear –about sensitivity analysis. It was done or not! If not why not?
- Under land use – only "forest and cropland" has been considered – is it because of data availability?
- AHP is a popular method for making expert judgement; however, it can be very complex and time consuming to communicate with the expert respondents; it might be interesting for the readers to learn from your experiences. Moreover, what are the criteria for being an expert for answering your AHP Matrix?
- Some of the discussion may help – why not other MCA approaches was considered like ANP….
- There are number of literatures has been already included – it might be relevant to look more on:
- https://www.sciencedirect.com/science/article/abs/pii/S2212420915301023
- https://www.researchgate.net/profile/Asad_Asadzadeh/publication/271065059_Assessing_Site_Selection_of_New_Towns_Using_TOPSIS_Method_under_Entropy_Logic_A_Case_study_New_Towns_of_Tehran_Metropolitan_Region_TMR/links/5655a88208ae4988a7b0de9e.pdf

- The presentation of the results needs to be improved further. For example, the cartographic presentation e.g. color combination may rethink for better visualization of results. For example, following presentation of the whole study area map, it will be nice to see some high resolution map by zoom on some specific critical area for a close look on the output.

- The discussion might be highlighted about the combination experience of multiple data sources, what are the major challenges. So far you have been using open data and automated workflow!! How about transferability and reproducibility of your proposed approach for countries that are having similar context and challenges.

- The conclusion may summarize the significant results and contributions (i.e. in bullet points).

---

## Author Comment (AC1) · 5 Jan 2021

Paper ID: https://doi.org/10.5194/nhess-2020-195-RC1, 2020 Paper Title: Integrated evaluation of water-related disasters using the analytical hierarchy process under land use change and climate change issues in Laos

We wish to thank you all for your constructive comments in this round of review. Your comments provide valuable insights to refine its contents and analysis. In this document, we try to address the issues raised as best as possible

Q:Line 19: a comma is essential after country

A:The comma has been added on line 19

Q:line 22: instead of 'can lead to' you may change to 'can increase'

A: We revised it as your comment.

Q: It seems that the current version lacks of international significance of this work. Hence think they may consider the following works to improve its readership. Furthermore, authors reviewed existing works but missed many in the area https://royalsocietypublishing.org/doi/pdf/10.1098/rsos.191957 https://www.sciencedirect.com/science/article/pii/S2212420920312632 https://www.sciencedirect.com/science/article/pii/S0264837720305470 https://www.nature.com/articles/s41598-020-69233-2

A:We agree with the referee comment. The additional references are important for our work. Therefore, the literature recommended have been added in the introduction section on • line 34-36 In addition, based on Adnan (2020) study on land use/land cover change and flood hazard on poverty in Bangladesh. At the end of their study, they argue that disorganized planning for land use is can increasing flood and poverty • line 40-45 Shah (2020) simulate for surface water under different climate change scenarios using set of regional circulation model (RCM) and soil and water assessment tool (SWAT) model for mid-century (2040-2070) and late century (2071-2100). The result of SWAT under future scenarios shows increase in steam flow for mid to late 21th century. However, the increase of steam flow for mid-century was a bit higher compare to late century due to the increase of temperature impact to snowfall and accumulation. • line 82-87 Yousefi (2020) produced multi hazard risk map in mountainous area using machine learning such as support vector machine, boosted regression tree, and generalized linear model to find the best model for each hazard and then create an integrated multi hazard in ArcGIS by adding each hazard together. Not only the technical capabilities of multi hazard map have to be consider but also the design of

information provided on multi hazard map have play as important role for end user's preferences(Dallo et al., 2020).

Q:This requires serious attention Section 3.2: What do you mean by expressions in lines 108-109? Unclear

A: This sentence was not clear. We revised it as follows. The overland flow has two runoff processes, which are surface flow and subsurface flow, and these flows are connected by infiltration process. More detailed information is available from Phrakonkham(2019) as shown in the main text.

Q: What was the resolution of DEM and what was the vertical accuracy of the model?

A: The DEM on the model is 1km x 1km made from the original data with a spatial resolution 90 m x 90 m for the distributed model. Shuttle Radar Topographic Mission (SRTM) Digital Elevation Map (DEM) was used in this study and based on the 'The Shuttle Radar Topography Mission Data Validation and Applications Workshop, 2005' mentioned 6.2m as the absolute vertical accuracy.

Q: Clarify Section 3.6.1 This section requires describing the method clearly, how have you done this?

A: We agree with the referee comments about section 3.6.1 and have revised this section more clearly as below: We propose a hazard index, which is adapted from the relationship between velocity and flood depth (Sally et al., 2008). The index is used for the identification of dangerous area where most of adults are unable to stand in floodwater depth more than 1.5m and are unable to stand in flood water depth 0.5 m and velocity 2 m/s (Russo et al., 2014; U.S. Department of the Interior, 1988). The index is scaled from zero to one, with zero representing the lowest hazard and one representing the highest hazard, and is divided into four categories from small to very high hazard. A top table of Figure 1 shows these categories for velocity and flooded depth. Here the categories for flood depth were shown as a case of velocity 0 m/s as

one example in a middle table of Figure 1 and we obtained a relationship between flood depth and the hazard index on a bottom graph of Figure 1. This process providing to the hazard index was applied to the study area using velocity and flood depth by the numerical simulation.

Q: Existing texts do not support this Line 174: should be "we wanted to.."

A: The text in line 174 has been changed to "we wanted to..."

Q: Line 185: How they have been chosen? At random? Was there any ethical permission sought? What were the main elements of questionnaire?

A: We made the interview for all of experts of government offices in the field of hazards and risks. For the questionnaire we obtained ethical permission. The main elements of questionnaire in this study are to understand weighted values on important aspects used in making decision by experts for five criteria according to AHP process.

Q: Discussion section is not properly reflecting what are you trying to achieve relative to your objective(s). Specifically, analyze and interpret your findings with the aid of theory, show similarities, dissimilarities. How your finding(s) differs from theory? Existing works showed above may be of help.

A: These sentences have been added to the discussion part to improve the section in the text: Dankers and Feyen (2008) assessed the influence of climate change to future flood hazard in Europe. They concluded that discharge from many rivers will increase on both magnitude and frequency by the end of this century. However, a few rivers will decrease discharge especially in the northeast Europe region. Mirza et al (2011) indicated that climate change will highly influence the monsoon precipitation and will increase the frequency, magnitude and hazard of flood in south Asia such as India, Bangladesh and Pakistan. Bouwer (2010) considered future precipitation and socioeconomic change such as land use and asset value, and obtained the damage cost as future flood risk. He concluded that the climate change will increase the damage cost

of flood around 35 to 170% by 2040 in Netherland. Ciabatta (2016) investigated the impact on landslide in Italy using PRESSA model in central Italy. The model based on the relationship between rainfall and soil moisture condition (Ponziani et al., 2012). Although all these studies are similar to our estimation for each hazard, the evaluation unified these hazards have been not carried out for future projection. AHP is useful to integrate the different hazard and successfully proposes the hazard map, which is easy for people to understand the local hazard, using values provided by decision makers.

Q: Conclusion section is also need improvements. What are the limitations? What are the take-home message(s) of this work? Nothing is clear. As it currently stands, conclusion section is sketchy and does not lead to useful conclusion(s)

A: Some sentences have been added to the conclusion part to explain the limitation and take home messages of this work: There are some limitations of the AHP approach. The AHP approach supposes linear independence of alternatives and criteria. It is recommended for the future study to make a comparison between AHP and other multi criteria decision making approaches. AHP results are obtained from current conditions and are not guaranteed in the future. Longer analysis from now in Lao PDR is necessary to predict more reliable future situation. In addition, a hazard map with this study resolution cannot explain it in smaller scale areas. DEM with higher resolution will be required for understanding of local hazard.

Q: Reduce number of maps in the work, show only crucial ones and the rest can go into Sup Info

A: Figure 5 to 7 have been moved to supplements.

Please also note the supplement to this comment:
https://nhess.copernicus.org/preprints/nhess-2020-195/nhess-2020-195-AC1-
supplement.pdf

2020-195, 2020.

---

## Author Comment (AC2) · 5 Jan 2021

Paper ID: https://doi.org/10.5194/nhess-2020-195-RC2, 2020 Paper Title: Integrated evaluation of water-related disasters using the analytical hierarchy process under land use change and climate change issues in Laos

We wish to thank you all for your constructive comments in this round of review. Your comments provide valuable insights to refine its contents and analysis. In this document, we try to address the issues raised as best as possible

Q:The title seems to be long and less declarative. Changing to "Mapping" might be a good fit than "Evaluation"

A:The title now been amended as suggestion

Q:The abstract may be improved – highlighting generalization of results and limitations of this study approach

A:For the limitation of this study, we added it in the conclusion part and added the limitation briefly from the part as follows. The integrated hazard maps can pinpoint the dangerous area through the whole country and the map can be used as primarily data for selected future development area. There are some limitations of the AHP methodology, which supposes linear independence of alternatives and criteria.. The conclusions was added by the following sentences. There are some limitations of the AHP approach. AHP approach supposes linear independence of alternatives and criteria. It is recommended for the future study to make a comparison between AHP and other multi criteria decision making approach. Moreover, for modelling the hazard map in smaller area, topographic information should have higher resolution for better understanding the hazard by local people

Q:Introduction may restructure – pushing the facts about the case study (national) a bit later, better say something at the very beginning about international facts as a motivation of this study

A:We agree with the referee about restructure of introduction. Therefore, we have added international facts in the beginning of introduction section as bellow: Now a day, natural disasters take a few thousand people life around the world and lose about a hundred billion USD every year (UNISDR, 2015). Additionally, Dilley (2005) has analyzed that about 700 million people and about 100 million people in the world are affected by at least two hazards and three or more hazards, respectively

Q:It is understandable, the author is introducing the AHP as a method in the introduction; however, the objective comes very late. Here it may help to be short, but specific to the research gap. Anyway, AHP related discussion are also part in the method section.

A:The text in introduction section now been amended as suggestion

Q:In the methodology, it remain unclear –about sensitivity analysis. It was done or not! If not why not?

A:In this study we did not apply sensitivity analysis because parameters were calibrated by a trail and error method comparing with observation data

Q:Under land use – only "forest and cropland" has been considered – is it because of data availability?

A:Reviewer's comment is right. We can considered only "forest and cropland" on land use according to the Laos national report (Laos national report, 2012)

Q:AHP is a popular method for making expert judgement; however, it can be very complex and time consuming to communicate with the expert respondents; it might be interesting for the readers to learn from your experiences. Moreover, what are the criteria for being an expert for answering your AHP Matrix?

A:All experts for the questionnaire are working in the administrative divisions in field of our concerned hazards and risk and have experienced the disaster survey and the communication to local people.

Q:Some of the discussion may help – why not other MCA approaches was considered like ANP….

A:We explained why we choose AHP method instead of other MCA methods, from line 103 to line 117 in the introduction section.

Q:There are number of literatures has been already included – it might be relevant to look more on: https://www.sciencedirect.com/science/article/abs/pii/S2212420915301023

https://www.researchgate.net/profile/Asad_Asadzadeh/publication/271065059_Assessing_S

A:New citations have now been updated to introduction section from • line 78 to line 80 For instance, Asadzadeh (2014) used TOPSIS model to find the solution in urban and regional planning issues and evaluated for site selection of new towns. • line 97 to line 99. For example, Asadzadeh (2015) used factor analysis with ANP (F'ANP) to construct a new set of parameters for earthquake resilience indicator.

Q:The presentation of the results needs to be improved further. For example, the cartographic presentation e.g. color combination may rethink for better visualization of results. For example, following presentation of the whole study area map, it will be nice to see some high resolution map by zoom on some specific critical area for a close look on the output.

A:New figures for the critical areas have now been amend as referee suggestion.

Q:The discussion might be highlighted about the combination experience of multiple data sources, what are the major challenges. So far you have been using open data and automated workflow!! How about transferability and reproducibility of your proposed approach for countries that are having similar context and challenges.

A:We appreciate too much for your suggestion. The text in discussion section now been revised to provide more detail our challenges. Ungaged areas have difficulty of analysis. Therefore, multiple open data sources were used in this study. Also poor observed data for disasters makes it difficult to calibrate and validate the results. It will be necessary to transfer qualitative data to quantitative data. The proposed approach in this research are not directly transferable and reproduceable in other countries that are having similar context because of the different in institutional and culture. Other countries can apply our proposed approach to produce their integrated hazard map but the weight priority of each hazard may depend on their expert judgements.

Q:The conclusion may summarize the significant results and contributions (i.e. in bullet

points).

A:The text in conclusion section is revised to summarize the significant results and contributions as follow: • The southern region has high and very high hazard areas comparing with the central region and the northern region. The Northern region has the lowest hazard area among three regions. • Total very high hazard area on the integrated hazard map with the anticipated change increases from 3.2% for RCP 2.6 to 3.27% for RCP 4.5 and up to 3.3% for RCP 8.5 in the near future (2010-2050) scenario. For the far future (2051-2099) scenario, the very high hazard area increases from 3.23% for RCP 2.6 to 3.52% for RCP 4.5 and up to 3.71% for RCP8.5

Please also note the supplement to this comment:
https://nhess.copernicus.org/preprints/nhess-2020-195/nhess-2020-195-AC2-supplement.zip

---

## Author Response (AR1)

**Paper ID: https://doi.org/10.5194/nhess-2020-195-RC1, 2020**

**Paper Title: Integrated evaluation of water-related disasters using the analytical hierarchy process under land use change and climate change issues in Laos**

We wish to thank you all for your constructive comments in this round of review. Your comments provide valuable insights to refine its contents and analysis. In this document, we try to address the issues raised as best as possible

**1st referee**

| | |
|---|---|
| Line 19: a comma is essential after country | The comma has been added on line 19. |
| line 22: instead of 'can lead to' you may change to 'can increase' | We revised it as your comment. |
| It seems that the current version lacks of international significance of this work. Hence think they may consider the following works to improve its readership. Furthermore, authors reviewed existing works but missed many in the area https://royalsocietypublishing.org/doi/pdf/10.1098/rsos.191957 https://www.sciencedirect.com/science/article/pii/S2212420920312632 https://www.sciencedirect.com/science/article/pii/S0264837720305470 https://www.nature.com/articles/s41598-020-69233-2 | We highly appreciate you for beneficial comments. We agree with the referee comment. The additional references are important for our work. Therefore, the literature recommended have been added in the introduction section on
 • line 34-36

 In addition, based on Adnan's study (2020) on land use/land cover change and flood hazard on poverty in Bangladesh. At the end of their study, they argue that disorganized planning for land use is increasing flood and poverty.
 • line 40-45

 Shah (2020) simulates surface water under different climate change scenarios using a set of regional circulation model (RCM) and soil and water assessment tool (SWAT) model for the mid-century (2040-2070) and the late century (2071-2100). The result of SWAT under future scenarios shows increase of steam flow for the mid to the late 21th century. However, the increase of steam flow for the mid-century was slightly higher compared |

| | with the late century due to the increase of temperature impact on snowfall and its accumulation.
• line 83-88

Yousefi (2020) produced a multi hazard risk map in a mountainous area using machine learning such as support vector machine, boosted regression tree, and generalized linear model to find the best model for each hazard. |
|---|---|
| This requires serious attention Section 3.2: What do you mean by expressions in lines 108-109? Unclear | This sentence was not clear. We revised it as follows.

The overland flow has two runoff processes, which are surface flow and subsurface flow, and these flows are connected by infiltration process.

More detailed information is available from Phrakonkham(2019) as shown in the main text. |
| What was the resolution of DEM and what was the vertical accuracy of the model? | The DEM on the model is 1km x 1km made from the original data with a spatial resolution 90 m x 90 m for the distributed model. Shuttle Radar Topographic Mission (SRTM) Digital Elevation Map (DEM) was used in this study and based on the 'The Shuttle Radar Topography Mission Data Validation and Applications Workshop, 2005' mentioned 6.2m as the absolute vertical accuracy. |
| Clarify Section 3.6.1 This section requires describing the method clearly, how have you done this? | We agree with the referee comments about section 3.6.1 and have revised this section more clearly as below:

We propose a hazard index, which is adapted from the relationship between velocity and flood depth (Sally et al., 2008). The index is used for the identification of hazard area where most of adults are unable to stand in floodwater depth more than 1.5m and are unable to stand in flood water depth 0.5 m and velocity 2 m/s (Russo et al., 2014; U.S. Department of the Interior, 1988). The index is scaled from zero to one, with zero representing the lowest hazard and one representing the highest hazard, and is divided into four categories from small to very high hazard. A top table of Figure 1 shows these categories |

| | for velocity and flooded depth. Here the categories for flood depth were shown as a case of velocity 0 m/s as one example in a middle table of Figure 1 and we obtained a relationship between flood depth and the hazard index on a bottom graph of Figure 1. This process providing to the hazard index was applied to the study area using velocity and flood depth by the numerical simulation. |
|---|---|
| Existing texts do not support this Line 174: should be "we wanted to.." | The text in line 174 has been changed to "we wanted to…" |
| Line 185: How they have been chosen? At random? Was there any ethical permission sought? What were the main elements of questionnaire? | We made the interview for all of experts of government offices in the field of hazards and risks. For the questionnaire we obtained ethical permission. The main elements of questionnaire in this study are to understand weighted values on important aspects used in making decision by experts for five criteria according to AHP process. |
| Discussion section is not properly reflecting what are you trying to achieve relative to your objective(s). Specifically, analyse and interpret your findings with the aid of theory, show similarities, dissimilarities. How your finding(s) differs from theory? Existing works showed above may be of help. | These sentences have been added to the discussion part to improve the section in the text:

Dankers and Feyen (2008) assessed the influence of climate change to future flood hazard in Europe. They concluded that discharge from many rivers will increase on both magnitude and frequency by the end of this century. However, a few rivers will decrease discharge especially in the northeast Europe region. Mirza et al (2011) indicated that climate change will highly influence the monsoon precipitation and will increase the frequency, magnitude and hazard of flood in south Asia such as India, Bangladesh and Pakistan. Bouwer (2010) considered future precipitation and socioeconomic change such as land use and asset value, and obtained the damage cost as future flood risk. He concluded that the climate change will increase the damage cost of flood around 35 to 170% by 2040 in Netherland. Ciabatta (2016) investigated the impact on landslide in Italy using PRESSA model in central Italy. The model based on the relationship |

| | between rainfall and soil moisture condition (Ponziani et al., 2012). Although all these studies are similar to our estimation for each hazard, the evaluation unified these hazards have been not carried out for future projection. AHP is useful to integrate the different hazard and successfully proposes the hazard map, which is easy for people to understand the local hazard, using values provided by decision makers. |
|---|---|
| Conclusion section is also need improvements. What are the limitations? What are the take-home message(s) of this work? Nothing is clear. As it currently stands, conclusion section is sketchy and does not lead to useful conclusion(s) | Some sentences have been added to the conclusion part to explain the limitation and take-home messages of this work:

There are some limitations of the AHP approach. The AHP approach supposes linear independence of alternatives and criteria. It is recommended for the future study to make a comparison between AHP and other multi criteria decision making approaches. AHP results are obtained from current conditions and are not guaranteed in the future. Longer analysis from now in Lao PDR is necessary to predict more reliable future situation. In addition, a hazard map with this study resolution cannot explain it in smaller scale areas. DEM with higher resolution will be required for more understanding of local hazard. |
| Reduce number of maps in the work, show only crucial ones and the rest can go into Sup Info | Figure 5 to 7 have been moved to supplements. |

**2nd referee**

| | |
|---|---|
| The title seems to be long and less declarative. Changing to "Mapping" might be a good fit than "Evaluation" | The title now been amended as suggestion. |
| The abstract may be improved – highlighting generalization of results and limitations of this study approach | For the limitation of this study, we added it in the conclusion part and added the limitation briefly from the part as follows. The integrated hazard maps can pinpoint the dangerous area through the whole country and the map can be used as primarily data for selected future development area. There are some limitations of the AHP methodology, which supposes linear independence of alternatives and criteria.. The conclutions was added by the following sentences. There are some limitations of the AHP approach. AHP approach supposes linear independence of alternatives and criteria. It is recommended for the future study to make a comparison between AHP and other multi criteria decision making approach. Moreover, for modelling the hazard map in smaller area, topographic information should have higher resolution for better understanding the hazard by local people |
| Introduction may restructure – pushing the facts about the case study (national) a bit later, better say something at the very beginning about international facts as a motivation of this study | We agree with the referee about restructure of introduction. Therefore, we have added international facts in the beginning of introduction section as bellow: Now a day, natural disasters take a few thousand people life around the world and lose about a hundred billion USD every year (UNISDR, 2015). Additionally, Dilley (2005) has analyzed that about 700 million people and about 100 million people in the world are affected by at least two hazards and three or more hazards, respectively |
| It is understandable, the author is introducing the AHP as a method in the introduction; however, the objective comes very late. Here it may help to | The text in introductoin section now been amended as suggestion |

| | |
|---|---|
| be short, but specific to the research gap. Anyway, AHP related discussion are also part in the method section. | |
| In the methodology, it remain unclear –about sensitivity analysis. It was done or not! If not why not? | In this study we did not apply sensitivity analysis because parameters were calibrated by a trail and error method comparing with observation data |
| Under land use – only "forest and cropland" has been considered – is it because of data availability? | Reviewer's comment is right. We can considered only "forest and cropland" on land use according to the Laos national report (Laos national report, 2012) |
| AHP is a popular method for making expert judgement; however, it can be very complex and time consuming to communicate with the expert respondents; it might be interesting for the readers to learn from your experiences. Moreover, what are the criteria for being an expert for answering your AHP Matrix? | All experts for the questionnarire are working in the administrative divisions in field of our concerned hazards and risk and have experienced the disaster survey and the communication to local people. |
| Some of the discussion may help – why not other MCA approaches was considered like ANP…. | We explained why we choosed AHP method instead of other MCA methods, from line 101 to line 115 in the introduction section. |
| There are number of literatures has been already included – it might be relevant to look more on:

-
https://www.sciencedirect.com | We appreciate you for benefitial information. New citations have now been updated to introduction section from
• line 79 to line 81
For instance, Asadzadeh (2014) used TOPSIS model to find the solution in urban and regional planning issues and evaluated for site selection of new towns. |

| | |
|---|---|
| /science/article/abs/pii/S22124 20915301023

-

https://www.researchgate.net/profile/Asad_Asadzadeh/publication/271065059_Assessing_Site_Selection_of_New_Towns_Using_TOPSIS_Method_under_Entropy_Logic_A_Case_study_New_Towns_of_Tehran_Metropolitan_Region_TMR/links/5655a88208ae4988a7b0de9e.pdf | • line 97 to line 100.
For example, Asadzadeh (2015) used factor analysis with ANP (F'ANP) to construct a new set of parameters for earthquake resilience indicator. |
| The presentation of the results needs to be improved further. For example, the cartographic presentation e.g. color combination may rethink for better visualization of results. For example, following presentation of the whole study area map, it will be nice to see some high resolution map by zoom on some specific critical area for a close look on the output. | New figures for the critical areas have now been revised as referee suggestion. |
| The discussion might be highlighted about the combination experience of multiple data sources, what are the major challenges. So far you have been using open data and automated workflow!! How about transferability and | We appreciate too much for your suggestion.
The text in discussion section has been revised to provide more detail our challenges.

Ungaged areas have difficulty of analysis. Therefore, multiple open data sources were used in this study. Also poor observed data for disasters makes it difficult to calibrate and validate the results. It will be necessary to transfer qualitative |

| | |
|---|---|
| reproducibility of your proposed approach for countries that are having similar context and challenges. | data to quantitative data. The proposed approach in this research is not directly transferable and reproduceable in other countries that are having similar context because of the different in institutional and culture. Other countries can apply our proposed approach to produce their integrated hazard map but the weight priority of each hazard may depend on their expert judgements. |
| The conclusion may summarize the significant results and contributions (i.e. in bullet points). | The text in conclusion section is revised to summarize the significant results and contributions as follow:

• The southern region has high and very high hazard areas comparing with the central region and the northern region. The Northern region has the lowest hazard area among three regions.

• Total very high hazard area on the integrated hazard map with the anticipated change increases from 3.2% for RCP 2.6 to 3.27% for RCP 4.5 and up to 3.3% for RCP 8.5 in the near future (2010-2050) scenario. For the far future (2051-2099) scenario, the very high hazard area increases from 3.23% for RCP 2.6 to 3.52% for RCP 4.5 and up to 3.71% for RCP8.5 |